# Optimization of the Steam Explosion Pretreatment Effect on Total Flavonoids Content and Antioxidative Activity of Seabuckthom Pomace by Response Surface Methodology

**DOI:** 10.3390/molecules24010060

**Published:** 2018-12-24

**Authors:** Jianqiu Tu, Huiping Liu, Naxin Sun, Shaojuan Liu, Pei Chen

**Affiliations:** Key Laboratory of Food Nutrition and Safety, Ministry of Education, Department of Food Engineering and Biotechnology, Tianjin University of Science & Technology, Tianjin, 300457, China; tujianqiu1022@163.com (J.T.); snxtust@sina.com (N.S.); liu_fendouzheye@126.com (S.L.); chenpei689689@163.com (P.C.)

**Keywords:** steam explosion, total flavonoids, antioxidant capacity, seabuckthom pomace

## Abstract

Steam explosion pretreatment was conducted on seabuckthom pomace. Response surface methodology was used to optimize the treatment conditions of steam explosion, including steam pressure, duration and particle size. After this, the content of total flavonoids and the antioxidant capacity of total flavonoids were investigated. Results showed that when the steam pressure was 2.0 MPa, duration was 88 s and a sieving mesh size was 60, the total flavonoids content in seabuckthorm reached a maximum of 24.74 ± 0.71 mg CAE/g, an increase of 246% compared with that without steam explosion treatment (7.14 ± 0.42 mg CAE/g). Also, DPPH and ·OH free radical scavenging ability showed significant improvement, with an IC_50_ decrease to 13.53 μg/mL and 4.32 μg/mL, respectively, far lower than that in original samples. Through the scanning electron microscope, the surface of seabuckthom pomace after steam explosion was crinkled, curly, and holey. Our study showed that the content of total flavonoids in seabuckthom pomace could be obviously promoted and the antioxidant capacity of total flavonoids also improved significantly, after applying steam explosion pretreatment to seabuckthom pomace, making this approach meaningful for the reuse of seabuckthom pomace resources.

## 1. Introduction

Seabuckthom, a perennial deciduous shrub or small arbor, is a kind of small berry plant that favors sunlight, resists water and drought, and enjoys vigorous growth. It is mostly orange or orange red and is widely planted in temperate zones in Europe and Asia [1]. China, is a country rich in seabuckthom resources. It mainly planted in the western region, with a total area of some 18 million hectares. The widespread planting of seabuckthom can not only prevent wind, fix sand, maintain water and soil, and ultimately facilitate the protection of the ecological environment, but also more importantly, it possesses the characteristic of being useful as a medicine and food owing to its rich content of nutrients like vitamins, sugar, proteins, amino acids and microelements, as well as biological active substrates. Therefore, it has attracted widespread attention in the research and development of healthcare products and foods [2,3,4].

Seabuckthom pomace (SP), a by-product after squeezing seabuckthorn juice, is mostly discarded except for a small proportion used as animal feed. After bacteriolysis, this discarded matter will rot and produce unpleasant smells, which not only pollute the environment but also causes a huge waste of resources [5]. It was found that there are still abundant total flavonoids in seabuckthorn pomace, total flavonoids (TFs), a generic term for a mixture of natural flavonoids, can mainly be used for anti-tumor and cardiovascular disease treatment [6]. Clinical studies have also proved that it also has ability to protect the sound organs and has antioxidant functions like reduction of blood fat, expansion of vessels, lowering of blood pressure, and liver protection as well as reduction of related enzymes [7]. Moreover, it has potential officinal value, being a kind of common myocardium protection substance and a top option for patients with cardiovascular diseases and high altitude workers. As a kind of drug and healthcare product, it has been accepted for its wide application and development prospects, so many studies have explored the TFs in seabuckthom fruits. Hou et al. extracted TFs from seabuckthom fruits and analyzed their regulatory effects on cellular immunity [8]. Jiang et al. extracted TFs from seabuckthom fruits and used them for treating lipopolysaccharide- induced inflammation [9].

Steam explosion (SE) technology is a biomass pretreatment means developed by Mason in 1928, in which s raw material is heated to 160 to 260 °C by saturated water steam at a certain pressure for a duration of several seconds to minutes, and then the pressure is suddenly reduced to atmospheric pressure. Its rationale lies in exposing the raw material to a high-temperature and high-pressure environment where it would be expanded due to over-heating liquid with steam filling the gaps; when the pressure is removed instantly (within 0.00875 s), then the over-heated intracellular solution in the gaps will be vaporized very fast so that the instantly expanding volume will cause a “cellular explosion”, forming many holes in the cell walls which will allow molecular substances to be released from cells [9,10,11]. Steam explosion methodology has been favored by researchers due to its low cost, and energy consumption and zero pollution, and has been widely applied in the pretreatment of raw materials for effective utilization [12,13,14]. Because of its ability to destroy cell walls, steam explosion has seen further development in the reuse of by-products in the food processing industry in recent years. For example, Wang et al. used steam explosion to treat straws to improve the extraction of xylitol [15]; Wang et al. used a combined steam explosion and acid extraction technology to treat orange peels to significantly improve soluble food fiber extraction [15]. However, there has been no report about the effect of steam explosion on the TFs content and the antioxidant capacity of TFs in seabuckthom pomace (SP). In our study, for the first time, we use the steam explosion technology to treat SP. The steam explosion conditions were optimized by using response surface methodology (RSM). The content and antioxidant capacity of TFs after pretreatment with steam explosion were investigated. This study could provide a scientific foundation for developing new natural antioxidants and reasonably utilizing SP resources.

## 2. Results and Discussion

### 2.1. Optimization of Steam Explosion Conditions for SP

#### 2.1.1. Statistical Analysis and Model Fitting

Response surface methodology (RSM) is an important method in the process of conditions optimization, which only needs fewer experiments to obtain statistically acceptable results. Table 1 presents the 17 response values (total flavonoids content) under different combinations of independent variables. Quadratic linear regression fitting and significance and variance analysis are applied to the data, and a second-order polynomial equation is given to express the dependent variable and response value. The equation was as follows:
Y = 24.74 + 0.88A − 0.53B − 0.68C − 3.94A^2^ − 4.28B^2^ − 1.92C^2^ − 1.05AB − 1.83AC − 0.015BC(1)

The results of variance analysis of the regression equation are presented in Table 2. The regression model is extremely significant (*p* < 0.0001 < 0.01), the lack of fit is not significant (*p* = 0.7247 > 0.05). The regression coefficient of the regression model equation is R^2^ = 0.992. Only less than 5% total variance is not explained by the model. These results indicate that the built model could soundly reflect the experiment data. The variance coefficient was 0.24, less than 0.5, indicating that the experiment data was reliable [16]. The one degree terms (A, B and C,) quadratic terms (A^2^, B^2^ and C^2^) and interaction terms (AB, AC) in regression model are extremely significant, while the interaction term BC was not, indicating that the influence of various experiment factors on the response value was not a simple linear relationship. The adjusted coefficient of determination (Adj R^2^) is 0.9821, demonstrating that the model represented the real relationship between the variables and that the adequacy of the responses is suitable.

#### 2.1.2. Optimization of Steam Explosion Pretreatment Conditions

The response surface and contour plots represent the interaction between two factors. The steeper the response surface, the denser the contours, the more obvious the interaction of the two factors is. Conversely, the interaction between the two factors is not obvious [17]. From Figure 1A,B, it can be known that when the particle size (C) is fixed at a sieving mesh size of 60, the TFs content increased first with steam pressure and duration, and then slowly dropped after reaching a certain level. This showed that under suitable pressure and duration conditions, steam pressure fully entered the cells, damaging the cell walls, and hence releasing the total flavonoids. However, as the steam pressure (A) and duration (B) kept increasing, excessive high-temperature and high-pressure would degrade the total flavonoids, and the flavonoids dissolved out of the cells would reaggregate to form insoluble substances [18]. From Figure 1C,D, it can be known that when duration (B) is fixed at 90 s, the TFs content increased first with steam pressure (A) and particle size (C), and then slowly dropped when a certain level was reached. This was because under suitable steam pressure and particle size conditions, the surface area of SP was increased, hence facilitating high-pressure steam to fully enter the cells [18]. From Figure 1E,F, it can be known that the interaction between duration and particle size had a slight influence upon the change of TFs content.

From the above discussions and analysis of the significance of quadratic polynomial model, the optimal conditions for pretreating SP with steam explosion were obtained as follows: steam pressure 2.01 MPa, duration 88 s, and sieving mesh size 54.5. Under this condition, the predicted TFs content was 24.91 mg CAE/g. Considering the actual operation conditions, the optimal conditions were slightly modified to steam pressure 2.0 MPa, duration 88 s, and sieving mesh size 60. Under these conditions, the TFs content would be 24.74 ± 0.71 mg CAE/g, with no significant difference from the predicted value, whereas for the SP without steam explosion pretreatment, the obtained content of TFs from the original SP was 7.14 ± 0.42 CAE/g. The results shown that the yield of TFs increased by 246% after steam explosion, proving that optimization of steam explosion pretreatment conditions for SP by the response surface methodology is feasible.

### 2.2. Microstructure Changes of SP after Steam Explosion under the SEM

As shown in Figure 2B, the SEM photographs shown that the SP without steam explosion pretreatment had a smooth and yellow surface, tight texture, and the particle size of SP was also very large. However, after steam explosion pretreatment the SP had a crinkled, curly and holey surface, (Figure 2A), and the bulk sample was divided into small sections. This was because in the pressure release, the steam penetrated into cells and tissues expanded and was instantly released at high speed, so the huge shearing force caused mechanical fracture of the cells [19].

### 2.3. The Effect of Steam Explosion on TFs Content and Antioxidation of SP

Free radicals can react with any molecule in active cells, rapidly causing pathological changes of tissue cells, various diseases and organic ageing. The antioxidative effect of flavonoids is due to the reaction between the flavonoid phenolic hydroxyl groups and free radicals, forming resonance stabilized free radicals and interrupting chain reactions [20].

#### 2.3.1. The Effect of Steam Explosion on the Ability of TFs to Scavenge DPPH Free Radical

After steam explosion, the DPPH free radical scavenging effects of SP TFs was measured. As shown in Figure 3, within the concentration range of 5–25 μg/mL, the DPPH free radical scavenging effect of TFs after steam explosion improved from 38% to 84% (black solid line); while that of TFs from the original SP improved from 21% to 63% (red broken line). 

This result indicates that after steam explosion, the content of TFs was improved, and the free radical scavenging ability was also increased. Moreover, the DPPH scavenging effect IC_50_ (half maximal inhibitory concentration) was 13.53 μg/mL, a 1.77-fold decline compared with the IC_50_ (37.46 μg/mL) of the original TFs.

#### 2.3.2. The Effect of Steam Explosion on the Ability of TFs to Scavenge·OH Free Radical

The ·OH free radical scavenging effects of TFs was measured after steam explosion. As shown in Figure 4, within the concentration range of 5–25 μg/mL, the ·OH free radical ability by TFs from SP after steam explosion was improved from 46% to 75.5% (solid line); while that of the original TFs improved from 28% to 53% (broken line). The results indicate that after steam explosion, with the increased content of TFsas, the free radical scavenging ability was also increased. Moreover, the ·OH scavenging activity IC_50_ was 4.32 μg/mL, a 78% decline compared with the IC_50_ (7.68 μg/mL) of the original TFs.

## 3. Materials and Methods

### 3.1. Materials

SP was obtained from Zhengzhou Dingjin Beverage Co., Ltd. (Zhengzhou, China); 1,1-diphenyl-2-picrylhydrazyl (DPPH) and catechin were from Sigma (St. Louis, MO, USA); absolute methanol, absolute ethyl alcohol, sodium nitrite, sodium hydrate, aluminium chloride, and salicylic acid were all analytical reagents obtained from local suppliers.

### 3.2. Sample Preparation

Cleaned and dried SP was put into a mincer for 20 s and passed through sieves with 20, 40, 60, 80, and 100 mesh. The seabuckthom fruit powders that passed through the different sieves were collected. The particle size can be represented by sieving mesh sizes from 20 to 100.

### 3.3. Pretreatment of SP with Steam Explosion

SP with different particle sizes (sieving mesh sizes from 20 to 100) was weighed and put into the steam explosion jar, equipped with a piston. When a certain steam pressure (1–3 MPa) and duration (30–150 s) were applied, high-temperature and high-pressure gas entered the cylinder through the air valve. After reaching the set time, the inlet air valve was shut and the pressure released instantly (0.00875 s) to complete the explosion of the pomace. The SP samples were then collected and stored at −20 °C.

### 3.4. Response Surface Experimental Design

On the basis of single factor experiments and the Box-Behnken central combination experiment design rationale with three factors and three levesl [21], the TFs content extracted from SP after steam explosion was set as response value (Y). Steam pressure (A), pressure maintenance duration (B) and particle sizes (C) during the steam explosion pretreatment were taken as experimental factors. Then, response surface methodology was adopted to evaluate the influence of these factors on the response value, and thereafter these steam explosion conditions were optimized according to the TFs content.

### 3.5. Microstructure Observation of SP

Scanning electron microscopy (S-3400 Hitachi, Ltd., Tokyo, Japan) was applied to observe the microstructures of the SP. The samples were placed on specimen holders with double-sided Scotch tape and sputter-coated with gold (10 min, 2 mbar), the samples were transferred to the scanning electron microscope in due order at an accelerating voltage of 20.0 kV.

### 3.6. Preparation of SP Extract

The method of SP extraction was reported previously [22]. Briefly, SP (2.00 g) was put into 50 mL centrifuge tubes. Then, 70% methanol-water mixed solvent (40 mL) containing 0.1% hydrochloric acid was added for 30 min reaction in an ultrasonic cleaner, followed by 3 min centrifugation (8000 rpm/min). Then the supernatant was collected, and the sediment was dissolved again with 40 mL 70% methanol-water mixed solvent containing 0.1% hydrochloric acid. The above steps were repeated. The two time extracts were dried in a drying oven and stored at 4 °C.

### 3.7. TFs Content Measurement in Seabuckthom Fruit

The measurement of TFs was conducted as previously reported with slight modifications [23]. Catechin (0.5 mg) was weighed and dissolved with 70% methanol in a 10 mL volumetric flask. Then, 0.125, 0.250, 0.500, 1.000, and 2.000 mL of catechin solution were pipetted into 5 mL volumetric flasks, to prepare 0.0125–0.2000 mg/mL standards. SP extract and catechin standards (0.5 mL) were pipetted into a 10 mL centrifugal tube, and NaNO_2_ (2 mL, 2%, *m*/*v*) and AlCl_3_ (3 mL, 10%, *m*/*v*) solution was added. After 3 min reaction, 1 mol/L NaOH (4.5 mL) was added and allowed to react for 10 min. Then the optical absorbance was measured at 510 nm. The catechin concentration (C) was used as horizontal coordinate and the absorbance (A) as vertical coordinate to plot a standard curve, and the TFs content was calculated, which was represented as catechin equivalence (mg) per gram SP (mg CAE/g).

### 3.8. Measurement of DPPH Free Radical Scavenging Ability

The measurement of DPPH free radical scavenging ability was conducted as previously reported with slight modifications [24]. TFs extract (2 mL) from SP, 70% methanol (2 mL), 0.5 mmol/L newly prepared DPPH (1 mL in absolute ethanol) were mixed in a colorimetric tube for 30 min incubation in dark condition. Then absorbance (A) was measured at the 517 nm. For the control group (A_1_), absolute ethanol was used to replace the DPPH solution, and 70% methanol was used to replace the sample extract as a model control group (A_2_). The DPPH free radical scavenging ability by TFs was calculated using the following formula:
DPPH free radical scavenging ability % = [1 − (A − A_1_)]/A_2_ × 100(2)

### 3.9. Measurement of ·OH Free Radical Scavenging Ability

The measurement of ·OH free radical scavenging ability was conducted as previously reported with slight modifications [25]. TFs extract from SP (2 mL), 6 mmol/L FeSO_4_ (2 mL), and 6 mmol/L H_2_O_2_ (2 mL) were mixed and incubated for 10 min. Then, 6 mmol/L salicylic acid (2 mL) was added for 30 min incubation. Absorbance (A) was measured at 510 nm. An equal amount of distilled water was used to replace the salicylic acid (A_1_). In the blank control group, distilled water was used to replace the SP extract (A_2_). The ·OH free radical scavenging rate ability by flavonoids TFs was calculated using the following formula:
OH free radical scavenging ability % = [1 − (A − A_1_)]/A_2_ × 100(3)

## 4. Conclusions

Steam explosion pretreatment was applied to SP, to improve the content of TFs. The response surface methodology was adopted to optimize the steam explosion treatment conditions. Ultimately, the optimum conditions were obtained as follows: steam pressure 2.0 MPa, duration 88 s, and a sieving mesh sizes 60. Under this condition, the TFs content in seabuckthorm fruit extract was 24.74 ± 0.71 mg CAE/g, an improvement of 246% compared with that obtained from the original fluid (7.14 ± 0.42 CAE/g). After steam explosion, the TFs in SP have a higher ability of scavenging DPPH and ·OH free radical, the IC_50_ values of which were respectively 13.53 μg/mL and 4.32 μg/mL, significantly lower than that in the original fluid. This indicates that the SP after steam explosion can be used as a natural antioxidant.

## Figures and Tables

**Figure 1 molecules-24-00060-f001:**
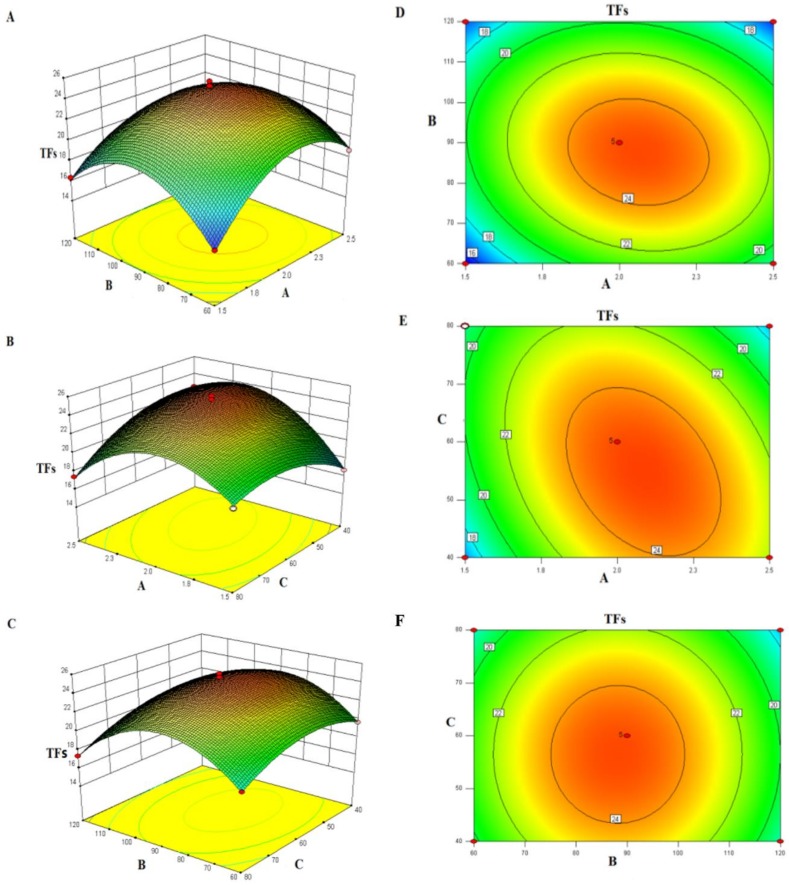
Response surface plot (**A**–**C**) and contour plot (**D**–**F**) showing the effects of the variables on content of TFs. The three independent variables set were steam pressure, duration and material diameter.

**Figure 2 molecules-24-00060-f002:**
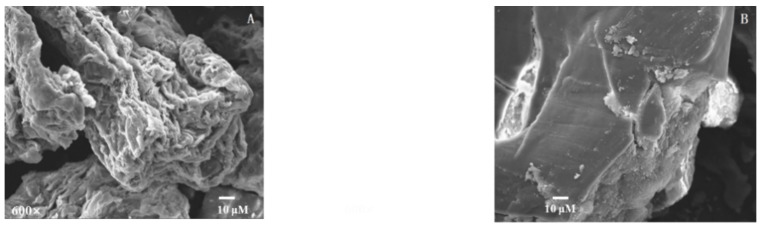
Scanning electron microcopy (SEM) images of SP (**A**: treated by steam explosion at 2.0 MPa, 88 s and 60 mesh untreated, **B**: original).

**Figure 3 molecules-24-00060-f003:**
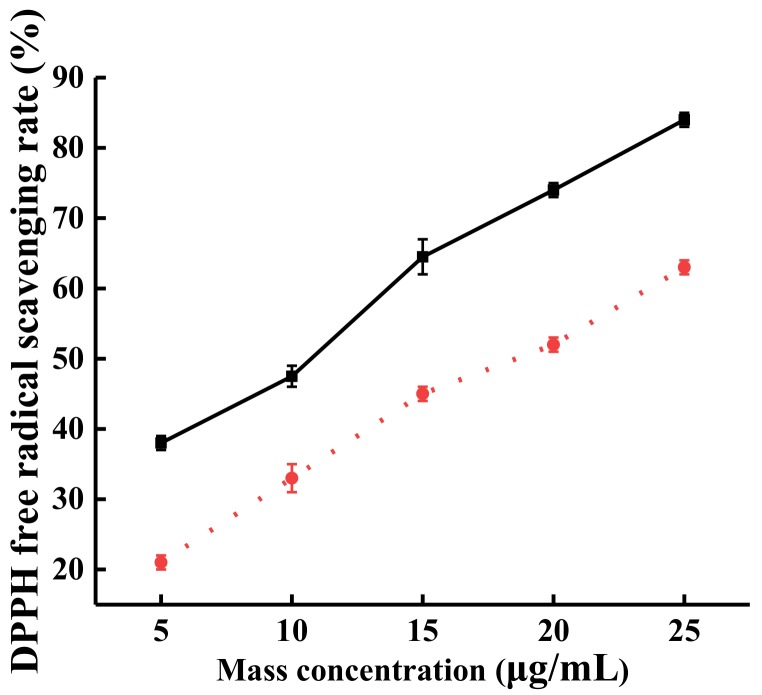
The effect of the TFs on the free radical scavenging of DPPH was obtained by steam explosion. Solid line: TFs after steam explosion, broken line: original TFs.

**Figure 4 molecules-24-00060-f004:**
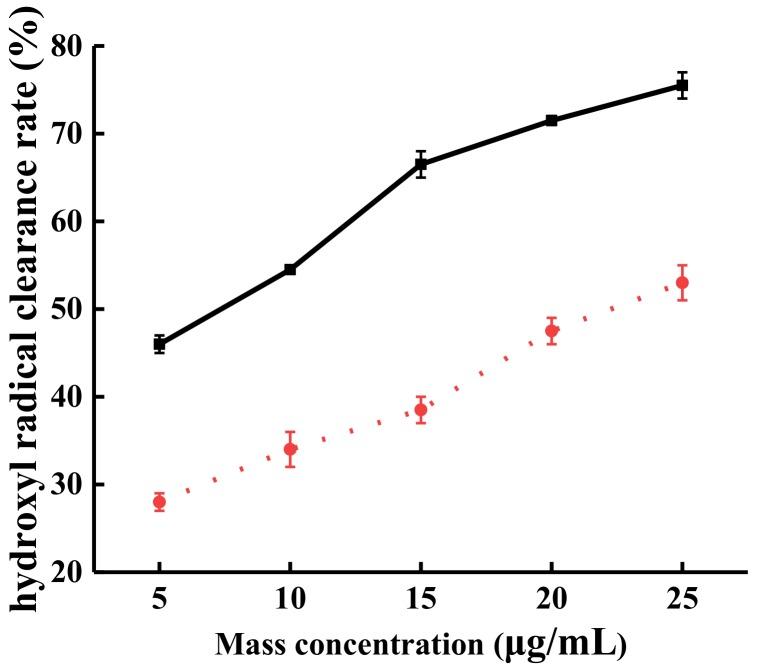
The effect of the extraction liquid on the free radical scavenging of hydroxyl was obtained by steam explosion. Solid line: TFs after steam explosion, broken line: original TFs.

**Table 1 molecules-24-00060-t001:** Central experiment design and results.

Runs	A Steam Pressure (MPa)	B Duration (s)	C Particle Size (mesh)	Y TFs (mg CAE/g)
1	2.5	60	60	18.75
2	2.5	120	60	15.62
3	2.0	120	40	18.65
4	2.0	90	60	24.24
5	2.0	90	60	24.51
6	2.0	90	60	25.52
7	2.0	120	80	17.36
8	2.0	60	40	19.69
9	1.5	90	80	18.89
10	2.0	90	60	24.34
11	2.5	90	80	17.44
12	1.5	120	60	16.39
13	1.5	60	60	15.34
14	2.0	90	60	25.09
15	2.5	90	40	22.54
16	2.0	60	80	18.46
17	1.5	90	60	16.67

**Table 2 molecules-24-00060-t002:** Regression equation variance analysis results.

Source	Sum of Squares	df	Mean Square	Coefficient	*F*-Value	*p*-Value
Model	203.99	9	22.67	24.74	98.76	<0.0001
A	6.23	1	6.23	0.88	27.15	0.0012
B	2.23	1	2.23	−0.53	9.7	0.0170
C	3.65	1	3.65	−0.68	15.88	0.0053
AB	4.37	1	4.37	−1.05	19.03	0.0033
AC	13.40	1	13.40	−1.83	58.37	0.0001
BC	9.000 × 10^−4^	1	9.000 × 10^−4^	−0.015	3.922 × 10^−3^	0.9518
A^2^	65.20	1	65.20	−3.94	284.08	<0.0001
B^2^	77.13	1	77.13	−4.28	336.08	<0.0001
C^2^	15.52	1	15.52	−1.92	67.63	<0.0001
Residual	1.61	7	0.23			
Lack of fit	0.41	3	0.14		0.46	0.7247
Pure error	1.19	4	0.30			
Cor total	205.60	16				
R^2^	0.992					
AdjR^2^	0.9821					
C.V. %	2.4					

*p* < 0.05 indicates statistical significance.

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
