# Peer review of "Optimization of the Steam Explosion Pretreatment Effect on Total Flavonoids Content and Antioxidative Activity of Seabuckthom Pomace by Response Surface Methodology"

_molecules, 2018, doi:10.3390/molecules24010060_

Round 1
Reviewer 1 Report
In this manuscript, the authors first introduced the steam explosion technique in pretreating the SP, using the response surface methodology to optimize the conditions. It should be a very useful protocol for fully obtaining the resources of SP.
Below are some comments:
1) In the whole manuscript, there is a lot of words, sentence mistakes such as IC50 should be subscript, TFS or TFs, the word "drug" at line 45, line 61 and 62 the alpretreated should be al pretreated, line 131 the word "through", line 173 TFSs, etc. Please double check the whole manuscript.
2) In Fig 3 and 4, what is the different color represent?
3) In the preparation of SP Extract protocol, the author add 0.1% hydrochloric acid into the solvent but did not neutralized it, would this acid condition affect the final analysis?
4) Is there any rule for designing the factor of the central experiment? In the table 1, the values of A, B and C seems changed randomly, For instance, in B factor, there was only 4 experiments for 60s but gave 9 experiments for 90s.
Author Response
Dear Editors and Reviewers:
Thank you for your letter and for the reviewers’ comments concerning our manuscript entitled "Optimization of Steam Explosion Pretreatment on Total Flavonoids Content and Antioxidation of Seabuckthom Pomace by Response Surface Methodology". (ID:383194). Those comments are all valuable and very helpful for revising and improving our paper, as well as the important guiding significance to our researches. We have studied comments carefully and have made correction which we hope meet with approval. Revised portion are marked in red in the paper. The main corrections in the paper and the responds to the reviewer’s comments areas flowing:
Responds to the reviewers comments:
Reviewer #1:
1. Response to comment: (In the whole manuscript, there is a lot of words, sentence mistakes)
Response: We are very sorry for our negligence of my mistakes, I had been modified in the article.
2. Response to comment: ( In Fig 3 and 4, what is the different color represent? )
Response: DPPH Free Radical Scavenging Effect by Seabuckthorm Fruit Extract before (red broken line) and after (black solid line) Steam Explosion
3. Response to comment:(In the preparation of SP Extract protocol, the author add 0.1% hydrochloric acid into the solvent but did not neutralized it, would this acid condition affect the final analysis?)
Response:we just use the 70% methanol-water mixed solvent containing 0.1% hydrochloric acid wash the sediment. And the sediment was washed by water, finally. So it will not affect the final analysis.
4. Response to comment:(Is there any rule for designing the factor of the central experiment? In the table 1, the values of A, B and C seems changed randomly, For instance, in B factor, there was only 4 experiments for 60s but gave 9 experiments for 90s.)
Response:According to the central combination experiment, the time chosen as 90s is the central experiment, so it must be given more experiments. The experiments were calculated by the software named Design 8.0.
Special thanks to you for your good comments.
We tried our best to improve the manuscript and made some changes in the manuscript. These changes will not influence the content and framework of the paper. And here we did not list the changes but marked in red in revised paper.
We appreciate for Editors/Reviewers’ warm work earnestly, and hope that the correction will meet with approval.
Once again, thank you very much for your comments and suggestions.
Yours sincerely,
Dr.Tu
Corresponding author:
Name: Huiping Liu
E-mail: liuhuiping111@163.com
Reviewer 2 Report
In the present study the others investigate by Response Surface Methodology the optimization of Steam Explosion Pretreatment on Total Flavonoids Content and Antioxidation of seabuckthom Pomace. This is a very interesting paper, the other hand this work presents majors corrections according to the following comments.
Comments to Authors:
- Why do you specifically talk about total flavonoids? Although in your work does not exist qualitative identification of these compounds.
- Page 4 figure 1, the figure lacks legend allowing a better legibility between the different schema A, B, C, D, E and F.
- Page 5 fig 2 show details of the Scanning electron microcopy (SEM) images of SP: these images lack details to exploit the difference between A and B.
- Page 5 Lin 128: You talk about "before and after Steam Explosion" but later in lin 133 and 144 you talk about "Extract after Steam Explosion".
- Page 5 Lin 135-136: Remove the repetition of the concentrations. Same remark for page 5 lin 146-147.
- Page 5 lin 140: I do not understand the declining percentage 177%.
- Page 5 lin 142: correct the title of figure 3. Add the legend at the level of the figure. Same remark for Figure 4 page 6.
- The entire Materials and Methods section lacks several details that are needed to report these experiments.
- Replace the Part Materials and Method before part Result and discussion.
Author Response
Dear Editors and Reviewers:
Thank you for your letter and for the reviewers’ comments concerning our manuscript entitled "Optimization of Steam Explosion Pretreatment on Total Flavonoids Content and Antioxidation of Seabuckthom Pomace by Response Surface Methodology". (ID:383194). Those comments are all valuable and very helpful for revising and improving our paper, as well as the important guiding significance to our researches. We have studied comments carefully and have made correction which we hope meet with approval. Revised portion are marked in red in the paper. The main corrections in the paper and the responds to the reviewer’s comments areas flowing:
Responds to the reviewers comments:
Reviewer #2:
1. Response to comment:(you specifically talk about total flavonoids? Although in your work does not exist qualitative identification of these compounds.)
Response: As we know, total flavonoids is a hybrid exist in Seabuckthorm Fruit. We can only obtain crude extract name as the total flavonoids by using the current method to prepare the total flavonoids.
2. Response to comment:(Page 4 figure 1, the figure lacks legend allowing a better legibility between the different schema A, B, C, D, E and F.)
Response: As shown in Figure 1, A and D, B and E, C and F represent response surface plot and contour plot, respectively. The higher the response surface is, the more dense the contour is, the more obvious the influence of interaction factors on dependent variables is. So the Figure 1 show that the influence of the interaction of every two factors between three factors on the TFs extraction rate.
3. Response to comment:(Page 5 fig 2 show details of the Scanning electron microcopy (SEM) images of SP: these images lack details to exploit the difference between A and B.)
Response: As Reviewer suggested that the line 123, "Fig. 2B, The photograph of the SEM shown that ", Line 124, "Their particle size is also very large.", Line 126, "Fig. 2A. The particle size was cut small section." was added.
4. Response to comment:(Page 5 Lin 128: You talk about "before and after Steam Explosion" but later in lin 133 and 144 you talk about "Extract after Steam Explosion".)
Response: We have made correction according to the Reviewer’s comments.Line 136-137, the statements of" Extract after Steam Explosion" were corrected as "before and after Steam Explosion"
Response to comment: (Page 5 Lin 135-136: Remove the repetition of the concentrations. Same remark for page 5 line 146-147.)
Response: It is really true as Reviewer suggested that the line 139-140 and line 151-152, "of 5 μg/mL, 10 μg/mL, 15 μg/mL, 20 μg/mL, 25 μg/mL " was deleted.
5. Response to comment:(Page 5 lin 140: I do not understand the declining percentage 177%.)
Response: We are very sorry for our incorrect writing the statements of"177%." were corrected as "177-fold compared with" in line 144.
6. Response to comment:(Page 5 lin 142: correct the title of figure 3. Add the legend at the level of the figure. Same remark for Figure 4 page 6.)
Response: As Reviewer suggested that "2.6.2. HE Staining." was deleted and "solid line: after steam explosion, broken line: before steam explosion." was added of figure 3, the same as figure 4.
Special thanks to you for your good comments.
We tried our best to improve the manuscript and made some changes in the manuscript. These changes will not influence the content and framework of the paper. And here we did not list the changes but marked in red in revised paper.
We appreciate for Editors/Reviewers’ warm work earnestly, and hope that the correction will meet with approval.
Once again, thank you very much for your comments and suggestions.
Yours sincerely,
Dr.Tu
Corresponding author:
Name: Huiping Liu
E-mail: liuhuiping111@163.com
Reviewer 3 Report
Dear authors,
the manuscript needs a large revision before the further evaluation.Moreover, I thing that this paper would be more suitable for some technology - related journal.
The language of manuscript is not clear, I find many sentences to be reformulated and typos to be corrected. The format of text should be changed to be more clear to sound scientific. The format of equotations is not easy to read, the captions of figures are not informative enough etc..
Round 2
Reviewer 2 Report
The authors have made all the necessary corrections. They have considerably improved their manuscript. I recommend its publication